# A pilot multicentre cluster randomised trial to compare the effect of trauma life support training programmes on patient and provider outcomes

Martin Gerdin Wärnberg ,[1,2] Johanna Berg ,[1,3] Prashant Bhandarkar,[4,5] Anirban Chatterjee,[6] Shamita Chatterjee,[7] Chintamani Chintamani,[8] Li Felländer-Tsai,[9,10] Anita Gadgil,[5] Geeta Ghag,[11] Marie Hasselberg,[1] Catherine Juillard,[12] Monty Khajanchi,[13] Deepa Kizhakke Veetil,[14] Vineet Kumar,[15] Debabrata Kundu,[16] Anurag Mishra,[17] Priti Patil,[5,18] Nobhojit Roy ,[1,5] Amit Roy,[19] Siddarth David,[1,20] Rajdeep Singh,[17] Harris Solomon,[21] Kapil Dev Soni,[22] Lovisa Strömmer,[23,24] Megha Tandon,[8] Trauma life support training Effectiveness Research Network (TERN) collaborators

For numbered affiliations see end of article.

**Correspondence to**
Dr Martin Gerdin Wärnberg;
martin.gerdin@ki.se

## ABSTRACT

**Introduction** Trauma accounts for nearly 10% of the global burden of disease. Several trauma life support programmes aim to improve trauma outcomes. There is no evidence from controlled trials to show the effect of these programmes on patient outcomes. We describe the protocol of a pilot study that aims to assess the feasibility of conducting a cluster randomised controlled trial comparing advanced trauma life support (ATLS) and primary trauma care (PTC) with standard care.

**Methods and analysis** We will pilot a pragmatic three-armed parallel, cluster randomised controlled trial in India, where neither of these programmes are routinely taught. We will recruit tertiary hospitals and include trauma patients and residents managing these patients. Two hospitals will be randomised to ATLS, two to PTC and two to standard care. The primary outcome will be all-cause mortality at 30 days from the time of arrival to the emergency department. Our secondary outcomes will include patient, provider and process measures. All outcomes except time-to-event outcomes will be measured both as final values as well as change from baseline. We will compare outcomes in three combinations of trial arms: ATLS versus PTC, ATLS versus standard care and PTC versus standard care using absolute and relative differences along with associated CIs. We will conduct subgroup analyses across the clinical subgroups men, women, blunt multisystem trauma, penetrating trauma, shock, severe traumatic brain injury and elderly. In parallel to the pilot study, we will conduct community consultations to inform the planning of the full-scale trial.

**Ethics and dissemination** We will apply for ethics approvals to the local institutional review board in each hospital. The protocol will be published to Clinical Trials Registry—India and ClinicalTrials.gov. The results will be published and the anonymised data and code for analysis will be released publicly.

---

### Strengths and limitations of this study

► Cluster randomised controlled trial comparing the effect of advanced trauma life support and primary trauma care and standard care on patient and provider outcomes.

► Prospective data collection with direct observations by dedicated project officers.

► Participating centres' heterogeneity may affect the study estimates and bias the results.

---

## INTRODUCTION

Trauma, defined as the clinical entity composed of physical injury and the body's associated response, causes 4.5 millions deaths every year.[1] Almost 10% of the global burden of disease is due to trauma and trauma is the top contributor to the burden of disease in children and adults aged 10–49 years.[2]

Trauma care is time sensitive and early management of life-threatening or limb-threatening condition is crucial. Several trauma life support training programmes have been developed to improve the early management of patients as they arrive at hospital by providing a structured framework to assessment and treatment.[3–5]

The proprietary advanced trauma life support (ATLS) is the most established trauma life support training programme and >1 million doctors in over 80 countries have been trained in the programme.[6] Uptake in low-income and middle-income countries (LMIC) has been slow, potentially due to high costs.[5]

The free primary trauma care (PTC) programme is the most widely spread alternative programme. The goal of PTC is to improve trauma care in LMIC.[7] Like ATLS, doctors in over 80 countries have been trained in PTC, and the programme has been endorsed by WHO, among other international organisations including several professional societies.[7]

Despite the widespread use of these training programmes, there are no controlled trials showing that they impact patient outcomes.[3–5] But there is level 1 evidence that these programmes improve provider skills and practices,[8 9] and observational data suggesting that they also improve patient outcomes.[10]

We will perform a pilot study that aims to assess the feasibility of conducting a cluster randomised controlled trial comparing ATLS and PTC with standard care. Recent methodological guidelines indicate that the design of efficient cluster randomised controlled trials requires data on probable or target effect sizes, proportion of participants with the outcome (if binary) and the intracluster correlation coefficient.[11] The objectives of this pilot study will be to:

► Estimate probable effect sizes on patient outcomes associated with ATLS and PTC compared with standard care, estimate the proportion of participants with the outcome (if binary) and estimate the intracluster correlation coefficient, as a basis for future sample size calculations.
► Assess the feasibility of recruiting participants and collecting data on primary and secondary outcomes, such as mortality, in-hospital complications, length of stay and quality of life.
► Assess how the effect sizes and directions of these effects of ATLS and PTC may differ across clinically important subgroups.

## METHODS
### Trial design
This study will pilot a pragmatic three-armed parallel, cluster randomised controlled trial, by the Trauma life support training Effectiveness Research Network (TERN, www.tern.network). There will be two intervention arms, ATLS and PTC training, and one control arm, standard care. We will collect data for 4 months in all three arms, first during a 1-month observation phase and then during a 3-month intervention phase (or continued observation in the control arm). This design will allow us to assess outcomes both as final values and as change from baseline. Our study is a pilot study because its objectives involve estimating quantities, such as the probable effect sizes, proportion of participants with the outcome (if binary) and the intracluster correlation coefficient, needed for the sample size calculations of a full-scale trial.[11] The full-scale trial will be planned regardless of the effect sizes identified in this pilot study. This pilot study will also establish how many participants that can be enrolled, as well as likely dropout rates, and the feasibility of collecting primary and secondary outcomes.

### Study setting
We will conduct this pilot study in Indian tertiary hospitals, where neither ATLS, PTC nor any other trauma life support training programme is routinely taught. India is the world's second most populous country and has 20% of the world's trauma deaths. The trauma system is still developing, with limited prehospital care, and the in hospital trauma mortality as well as the proportion of preventable deaths remain high. Lack of standard trauma training for healthcare providers, limited hospital resources, inadequate processes of care, overcrowding emergency departments are some of the factors that contribute to the high mortality and morbidity. During recent years, efforts have been made to improve hospital trauma care, through capacity building for trained trauma care providers, augmenting facilities and developing care protocols within the hospitals. Our pilot study is planned to start during 2022.

### Eligibility criteria for participants and clusters
There will be two groups of participants: patients and resident doctors.

#### Patient participants
Adults (aged 15 years or older) who present to the emergency department at participating hospitals with a history of trauma. History of trauma is here defined as having any of the external causes of morbidity and mortality listed in block V01-Y36, chapter XX of the International Classification of Disease version 10 (ICD-10) codebook as reason for presenting. We will explore intervention effects across the following clinical subgroups: men, women, blunt multisystem trauma, penetrating trauma, shock, severe traumatic brain injury and elderly, as defined by Hornor et al.[12] The consent form for patients are available in online supplemental material 1.

#### Resident doctor participants
Resident doctors doing their specialty training in surgery or emergency medicine, who manage trauma patients in the emergency department, and who are expected to remain in the participating hospitals for at least 1 year. To facilitate administration each clinical department is divided into units, which manages the outpatient department, emergency department, operating rooms, etc on different days each week. One or two, out of typically six, units' residents will be selected from each hospital. One unit consists of at least 3 faculty and 3–12 residents.

To be eligible, units should have a maximum of 25% of the doctors trained in either ATLS, PTC or similar training programmes before the start of the pilot (hospitals that have so far agreed to participate have no or single current residents trained in any programme). Those residents who have received training in the last 5 years will be considered as trained. The figure of 25% was decided through consensus in the research team, to balance feasibility and contamination of results. We will select the units by conducting a prior survey to ascertain

this criteria. Consent will be sought from the residents in each of the intervention groups before they undergo the ATLS or PTC training. The consent form for residents are available in online supplemental material 2. We will not ask for consent from residents at the units in the control hospitals as their practice will not be affected by this pilot study and we will not collect any personal identifiable data on them. This is in line with ethical regulations in the study setting.

### Clusters

Indian tertiary care hospitals that admit 400–800 adult patients with trauma each year. We randomise on the cluster (hospital) level to avoid contamination between intervention and control arms. To be eligible for inclusion, hospitals have to provide the following services round the clock: operation theatres, X-ray, CT and ultrasound facilities, and blood bank. In addition, the baseline admission rate should be >35 adult patients with major trauma per month.

### Interventions

In each intervention arm, one or two units', out of typically six, residents per hospital providing emergency care to trauma patients will be trained in either ATLS or PTC. For the purpose of this pilot study, we will target to train a minimum of 75% of residents in each unit. If residents drop out or change units after training but before data collection is completed, we will conduct additional training if needed to meet the 75% criterion. We will not train the units' faculty, as they are typically not involved in the initial management of trauma patients.

The ATLS training will be conducted in the nearest ATLS certified training centre in India according to the standard ATLS curriculum.[6] The PTC training will be arranged in hospitals randomised to the PTC arm, according to the standard PTC curriculum.[7] These courses will be conducted over a period of 2.5–3 days. The residents certified 'pass' will be considered as trained in respective courses.

The control group provides standard care with no intervention.

### Modifications

Both ATLS and PTC are standard training programmes with fixed curricula.[6 7] We will not modify the delivery or content of these programmes during this pilot.

### Adherence

The intervention is the training in either ATLS or PTC. Participants are required to adhere to, that is, participate in, the training, to be eligible for passing. We will not consider adherence to training contents during care delivery as adherence to the trial intervention, but rather as a provider-level outcome.

### Concomitant care
*Baseline training*

The care provided by all participating hospitals at baseline is based on the training curriculum formulated by The National Medical Council of India for postgraduation in general surgery.[13] Regarding trauma, these guidelines state that the student should:

1. have knowledge about response to trauma; burns: causes, prevention and management; wounds of scalp and its management; recognition, diagnosis and monitoring of patients with head injury, Glasgow coma scale;
2. be able to provide and coordinate emergency resuscitative measures in acute surgical situations including trauma;
3. choose, perform and interpret appropriate imaging in trauma—ultrasound focused abdominal sonography in trauma (FAST);
4. undergo advanced trauma and cardiac life support course (certified) before appearing in final examination;
5. undergo clinical posting in emergency and trauma;
6. present or discuss cases of blunt abdominal trauma.

Although training in an ATLS course is part of the curriculum, it is optional and not doing this training does not result in failure to obtain postgraduation completion.

*Standard of care*

At most medical colleges in India trauma patients present to the emergency department, where they are assessed by a doctor and referred to the surgical bay for further management. In the surgical bay, a second-year or third-year general surgery resident sees all the major trauma and provide the initial care, including initiating treatment and investigations. This resident informs the consultant on call who is generally an Assistant Professor. Most procedures like intercostal drainage, open wound suturing, intubation, etc would be done in the surgical resuscitation area, by the surgical resident.

Compared with other settings where a trauma team approach is adopted, nurses and other healthcare professionals are involved to a limited extent during the initial management. Their roles include assisting during intubation and other bedside procedures, charting the vitals (not recording) and giving injections. They also accompany the resident during transfers of serious patients.

After completing the assessment and starting initial resuscitation, the resident decides to send the patient for imaging (X-rays/FAST/CT scan) or to the operation room in consultation with or after assessment by the on-call consultant. A portable X-ray and an ultrasonography machine to conduct FAST may or may not be available in the surgical bay. The patients who are operated, managed conservatively, not intubated or with minor trauma will be sent to the surgical ward. Those who need increased monitoring or mechanical ventilation remain in the surgical bay or in the intensive care unit (ICU) depending on the availability of ICU beds. The further treatment continues in the respective

ward or ICU and patients are finally discharged from the ward.

## Outcomes

Our pilot study include both participant and feasibility outcomes. Prior to deciding on these participant outcomes, we searched the Core Outcome Measures in Effectiveness Trials Initiative's database but were unable to identify appropriate core outcome sets for our populations of participants.

The primary participant outcome will be all-cause mortality within 30 days from the time of arrival to the emergency department. The primary outcome and most secondary outcome will be assessed and compared both as final values and as change from baseline. All outcomes that pertain to the individual participant level are detailed in online supplemental material 3. We decided to include a large number of outcomes, including some more exploratory, so that we can test their feasibility and relevance. We may remove secondary participant outcomes during the course of the pilot study, if they prove to be too difficult to collect. If we remove outcomes, we will document the reasons for doing so.

We will also assess the following feasibility outcomes, which pertain both to overall study population as well as to the individual cluster level:

▶ Recruitment rate. For both patients and residents, this will equal the proportion of participants enrolled, out of the total number of eligible participants, over the course of the pilot study.

▶ Lost to follow-up rate. This will apply only to patients and equal the proportion of patients that do not complete 30-day follow-up, out of all enrolled patients, over the course of the pilot study.

▶ Pass rate. This will apply only to residents in the intervention arms and equal the proportion of residents that pass the training programme, out of the total number of trained residents, over the course of the pilot study.

▶ Missing data rate. This will apply to each outcome and variable and equal the proportion of missing data, over the course of the pilot study.

▶ Differences in distributions of observed and extracted data. This will apply to each outcome and variable and will compare the distributions of data collected by observations versus extracted from hospital records. For quantitative variables, this will be the difference in means, SD, medians, IQRs and ranges. For qualitative variables, this will be the differences in absolute counts and percentages, across categories.

## Participant timeline

### Patients

Patients will be screened for eligibility as they arrive at the emergency department. Eligible patients will be approached in the emergency department to consent to follow-up, if they are conscious. If they are unconscious, a patient representative will be approached to consent to follow-up. Once the patient is conscious, we will approach the patient to affirm the patient representative's consent. We will follow-up patients at discharge, at 24 hours after arrival at the emergency department, and at 30 days after arrival at the emergency department.

### Residents

Surgical units will be screened for eligibility once hospitals confirm their participation. All residents in eligible units will be approached to consent to training if their hospital is randomised to either of the intervention arms. Training will be conducted as soon as possible after the study starts. Resident participants will be followed up 30 days after training, if they are in the intervention arms, or 30 days after the study started, if they are in the control arm.

## Sample size

Given budget and time constraints, including the rotation of units in Indian hospitals (which often happen on a 6-month basis), the feasible data collection period is 4 months. Each of the units see two to four trauma patients per week. If we select a minimum of one unit per hospital then each hospital will enrol 8–16 patients per month and 32–64 patients during the 4 months of this pilot study. With a 20% attrition rate we expect each hospital to enrol 26–51 patients, coming to a total sample size of between 156 and 306 patients for this pilot study.

## Recruitment

To ensure adequate recruitment, we only approach hospitals with trauma volumes high enough to allow us to reach the sample size goals detailed above. Patients will be enrolled by a dedicated project officer as they arrive at the emergency department. The recruitment period will be 4 months. Recruitment will be monitored weekly through online conferences. No financial or non-financial incentives will be provided to trial investigators or participants for enrolment.

## Allocation

### Sequence generation

We will use simple randomisation to allocate sites to trial arms. We will prepare six sealed envelopes of which one representative from each pilot site will draw one. The content of the envelope will dictate what trial arm (ATLS, PTC or standard care) the hospital will be in. There will be two hospitals in each trial arm.

### Concealment mechanism

We will not conceal the sequence, see 'Sequence generation' section.

### Implementation

The random allocation sequence will be generated by the project's core group, who also enrol clusters. Patient participants will be included if they present during the project officers shift. Resident participants are enrolled if they are in the units selected for training. We will use

simple random sampling to select units if there are more than two eligible units in a hospital. For patient participants, consent for follow-up is sought after randomisation from patients or patient relatives as appropriate. For resident participants, consent is sought before randomisation. If residents in a unit decline to participate, so that the target of training 75% of residents in a given unit cannot be met, another unit will be selected for participation.

## Blinding

It will not be possible to blind investigators or participants to interventions. We will not blind the data analysts during this pilot, but we plan to blind the data analysts during the full-scale trial.

## Data collection

Data collection will start 1 month before the training is delivered, to establish a baseline. A variability of 3 months of the date when data collection is started between hospitals will be accepted. Each participating hospital will have a dedicated project officer to collect data. The project officers will have a masters in a health science field and should have experience in data collection.

Because participating residents are assigned designated days for trauma care for a period of 6 months, data will be collected during those particular days and shifts when these trained doctors are in the emergency department. The project officers will collect data both by observing the care delivered and by interviewing the participants, and by extracting data from hospital records.

Data collection will continue for a minimum of 3 months after training. The research officers will collect data of all patients, who present with trauma in the surgical bay during their duty hours. Those patients who are admitted will be followed up for complications and other in-hospital outcome measures, for example, length of stay. Patients who are not admitted will be followed up telephonically for mortality outcomes and quality of life outcomes. The follow-up period will be 30 days. The project officers will make at most three attempts to reach a participant or participant representative telephonically, after which the data will be recorded as missing.

The project officer will administer the study information and informed consent (consent will only be sought for data collection including follow-up) to the patient, or the patient's representative as appropriate, once the patient is stabilised. They will continue to collect data once they have received the consent.

Details of data of those patients/relatives not willing to give consent will be removed from the analysis. The number of patients who opt out from data collection will be collected, as well as limited data on their age and sex. Patients will be followed up in the ward regularly for the various outcome variables. They will also be followed up telephonically after they have been discharged.

## Variables

The project officers will collect data on demographics, time of injury to arrival at the participating hospital, time to recording vital signs, vital signs and times to and management details including imaging and surgery. Details of any injury sustained will be collected and coded using ICD-10 and the Abbreviated Injury Scale (AIS). For ICD-10, coders will undergo the WHO online ICD-10 training module and for AIS they will be accredited. Based on AIS, we will calculate the Injury Severity Score (ISS) and the New ISS. Online supplemental material 4 contains a full variable list, with definitions.

## Patient and public involvement

In this study, we will conduct community consultations to collect inputs from patients, their caregivers, patient groups and resident doctors to be used in the selection of outcome measures and implementation of the full-scale trial, following the Guidance for Reporting Involvement of Patients and the Public 2.[14]

During the pilot study, interviews will be conducted with postdischarge trauma patients and their caregivers to identify outcomes most relevant to them. These patients will be identified through the medical registers of the participating hospitals, contacted through telephone and after receiving their consent be interviewed as per their convenience. Their consent form is available as online supplemental material 5. Additionally, members from non-government organisations working with trauma patients and the hospital social service section will also be contacted for their views on contextual patient-centred outcomes for trauma patients. Their consent form is available as online supplemental material 6. For feasibility, these interviews will be held in each of the cities where the participating centres are located. The most common patient-centred outcomes reported across all the locations will be incorporated into the evaluation of the effects of the different training programmes and standardised care on patient outcomes.

Similarly, the inputs of resident doctor participants at each participating centre will be collected during the pilot study. A discussion and periodic surveys will be conducted to document any challenges or suggestions they may have in the scheduling or implementation of the training programmes. These inputs will be incorporated in the final study.

A summary of the findings of the study as well as their inputs will be shared with those who participated in the interviews and surveys. A meeting will be held with the patient participants, at each city, where the changes in the measured patient-centred outcomes would be presented to them. Another meeting will be held with the resident doctors at each hospital to present the confidence of the residents after being trained. Any suggestions and reflections from the participants during the meetings will be used as inputs for planning the final study.

## Data management

We will supply an online data collection tool, accessible only over a virtual private network, for each participating hospital to upload pseudonymised data to secure servers. Data validation techniques like restricted values or values of a specific range will be used to avoid ambiguous data entries and ensure the validity of the data. Ambiguous responses, data errors, if any, will be resolved after discussion with the core team during weekly meetings. An instruction manual or codebook for data variables will be prepared to ensure consistency in data entry. This manual will be referred to during the project data collection and variable descriptions are visible for each variable in the online data collection tool. Pseudonymised data will be stored at the centralised server. The data will be accessible by the project's principal investigator or by delegation of the project principal investigator only.

## Data monitoring

Weekly meetings with the core team and project officers will take place and for this meeting a data status report will be automatically generated highlighting missing data and number of patients awaiting follow-up. Cluster-specific interim analysis will take place after 2 months. The results of this will be presented to the core team, this team will decide if the pilot should be terminated. Although we will not have formal termination criteria because of the short duration of the study, reasons not to continue could include that the collection of key variables, such as mortality outcomes, is unfeasible or that patients are not consenting to be included in the data collection. A data monitoring committee will not be used in the pilot study due to its limited scope.

## Statistical methods

We will analyse all pilot data using descriptive statistics. Quantitative variables will be summarised as mean±SD, median, IQR and range. Qualitative variables will be presented as absolute numbers and percentages. Feasibility outcomes will be summarised both on the overall sample level as well as on the individual cluster level. We will use an empty generalised linear mixed model to estimate the intracluster correlation coefficient.

We will compare participant outcomes in three combinations of trial arms: ATLS versus PTC, ATLS versus standard care and PTC versus standard care. In each combination, we will compare both differences in final values and differences in change from baseline. For example, for the primary participant outcome of all-cause mortality within 30 days from the time of arrival to the emergency department, comparing ATLS versus PTC, we will compare both the difference in mortality between the ATLS and PTC arms as well as the difference in the change from baseline in mortality between the ATLS and PTC arms.

For the intervention arms, the change from baseline will be calculated as the difference between the 1-month period of data collection before the training was undertaken and the 3-month period after the training. For the control arm, the data collection period will be 4 months and the difference from baseline will be calculated as the difference between the first 1 month and the following 3 months.

Within each combination of trial arms, we will conduct subgroup analyses of men, women, blunt multisystem trauma, penetrating trauma, shock, severe traumatic brain injury and elderly. Table S7.1 in online supplemental material 7 shows which outcomes will be assessed in which subgroups, decided through consensus in the research team. We will further compare the results of all subgroups with the results in the whole cohort, and compare the results in the female subgroup with the male subgroup, and the results in the blunt multisystem trauma subgroup with the penetrating trauma subgroup. We are aware that the numbers in some of these subgroups are likely to be small, but we include them to help guide the formulation of the statistical analysis plan for the full-scale trial.

We will calculate both absolute and relative differences for each comparison, along with 75%, 85% and 95% CIs. We will use an empirical bootstrap procedure with 1000 draws to estimate these CIs. We will not perform any formal hypothesis tests during the analysis of this pilot's data.[15] We will also compare the data collected through observations and interviews with the data collected from hospital records, to assess the feasibility of collecting data from hospital records in the full-scale trial.

## ETHICS AND DISSEMINATION

We will apply for research ethics approval at local clusters in India to the local institutional review board committees. The protocol will be submitted for journal publication as well as to Clinical Trials Registry—India and ClinicalTrials.gov. Amendments to the protocol after this will be determined by the core research group and updated on Clinical Trials Registry—India and Clinical-Trials.gov. Substantial amendments, such as modifications to the eligibility criteria or outcomes will also be resubmitted to the journal. Declaration of interest will be submitted from all participating researchers both in the core team and at each site. The final anonymised dataset and code for analysis will be released publicly. The results will be submitted for publication in peer-reviewed open access journals. Authorship will follow the International Committee of Medical Journal Editors guidelines.

**Author affiliations**
[1]Department of Global Public Health, Karolinska Institutet, Stockholm, Sweden
[2]Function Perioperative Medicine and Intensive Care, Karolinska University Hospital, Stockholm, Sweden
[3]Emergency Medicine, Department of Internal and Emergency Medicine, Skåne University Hospital, Malmö, Sweden
[4]Tata Institute of Social Sciences School of Health Systems Studies, Deonar, Maharashtra, India
[5]World Health Organization Collaborating Center for Research in Surgical Care Delivery in Low-and-Middle Income Countries, Mumbai, India

⁶Department of Orthopaedic Sciences, Medica Superspecialty Hospital, Kolkata, India

⁷Department of Surgery, Seth Sukhlal Karnani Memorial Hospital, Kolkata, West Bengal, India

⁸Department of Surgery, Vardhman Mahavir Medical College and Safdarjung Hospital, New Delhi, Delhi, India

⁹Division of Orthopaedics and Biotechnology, Department of Clinical Science Intervention and Technology (CLINTEC), Karolinska Institutet, Stockholm, Sweden

¹⁰Department of Reconstructive Orthopedics, Karolinska University Hospital, Stockholm, Sweden

¹¹Department of Surgery, HBT Medical College and Dr R N Cooper Municipal General Hospital, Mumbai, India

¹²Division of General Surgery, Department of Surgery, David Geffen School of Medicine at UCLA, Los Angeles, UK

¹³Department of Surgery, Seth Gowardhandas Sunderdas Medical College and King Edward Memorial Hospital, Mumbai, India

¹⁴Department of Surgery, Manipal Hospital Dwarka, New Delhi, India

¹⁵Department of Surgery, Lokmanya Tilak Municipal Medical College and General Hospital, Mumbai, Maharashtra, India

¹⁶Department of Surgery, Medical College Kolkata, Kolkata, India

¹⁷Department of Surgery, Maulana Azad Medical College, New Delhi, Delhi, India

¹⁸Department of Statistics, Bhabha Atomic Research Centre Medical Division, Mumbai, Maharashtra, India

¹⁹Department of Surgery, Sir Nil Ratan Sircar Medical College & Hospital, Kolkata, India

²⁰Doctors For You, Mumbai, India

²¹Department of Cultural Anthropology and the Duke Global Health Institute, Duke University, Durham, North Carolina, USA

²²Critical Care, All India Institute of Medical Sciences, New Delhi, Delhi, India

²³Department of Surgery, Capio S:t Görans Hospital, Stockholm, Sweden

²⁴Division of Surgery, Department of Clinical Science, Intervention and Technology (CLINTEC), Karolinska Institutet, Stockholm, Sweden

**Twitter** Nobhojit Roy @#nobsroy

**Contributors** MGW conceived of the study. AG, AM, CJ, DKV, HS, JB, KDS, LF-T, LS, MH, MGW, MK, NR, PB, PP, RS, DS and VK contributed to the design of the study. DKV, KDS, MK and MGW drafted the first version of the protocol. AG, HS and DS drafted the first version of the patient and public involvement activities. JB and PP drafted the first versions of the data management sections and wrote the data management plan. PB and PP drafted the first versions of the statistical analysis section. AG, AM, CJ, DKV, HS, JB, KDS, LF-T, LS, MH, MGW, MK, NR, PB, PP, RS, SC, DS and VK contributed to the refinement of the protocol. AR, AC, C, DK, GG, MK, MT and VK are representatives of prospective participating hospitals.

**Funding** Doctors for You through grants awarded to Karolinska Institutet by the Swedish Research Council (grant number 2020-03779) and the Laerdal Foundation (grant number 2021-0048).

**Competing interests** Several authors are Advanced Trauma Life Support instructors.

**Patient and public involvement** Patients and/or the public were involved in the design, or conduct, or reporting, or dissemination plans of this research. Refer to the Methods section for further details.

**Patient consent for publication** Not applicable.

**Provenance and peer review** Not commissioned; externally peer reviewed.

**ORCID iDs**
Martin Gerdin Wärnberg http://orcid.org/0000-0001-6069-4794
Johanna Berg http://orcid.org/0000-0001-7553-7337
Nobhojit Roy http://orcid.org/0000-0003-2022-7416

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
