## [Reviewer comments · BMJ Open]

ARTICLE DETAILS

TITLE (PROVISIONAL)	A Pilot Multicenter Cluster Randomized Trial to Compare The Effect of Trauma Life Support Training Programs on Patient and Provider Outcomes
AUTHORS	Gerdin Wärnberg, Martin; Berg, Johanna; Bhandarkar, Prashant; Chatterjee, Anirban; Chatterjee, Shamita; Chintamani, Chintamani; Felländer-Tsai, Li; Gadgil, Anita; Ghag, Geeta; Hasselberg, Marie; Juillard, Catherine; Khajanchi, Monty; Kizhakke Veetil, Deepa; Kumar, Vineet; Kundu, Debabrata; Mishra, Anurag; Patil, Priti; Roy, Nobhojit; Roy, Amit; Siddarth, David; Singh, Rajdeep; Solomon, Harris; Soni, Kapil; Strömmer, Lovisa; Tandon, Megha

VERSION 1 – REVIEW

REVIEWER	Tisherman, Samuel A. University of Maryland School of Medicine
REVIEW RETURNED	26-Nov-2021

GENERAL COMMENTS	General comments The authors have proposed a study of the effect of the Advanced Trauma Life Support (ATLS) course or the Primary Trauma Course (PTC) on patient outcomes. The impact of these courses has been difficult to test. Since these courses have not been routinely taught in India, this study is uniquely positioned to test this impact. Overall, the study is well-described and well-designed given the many difficulties and potentially confounding factors in such a study. In general, since the hospitals will likely have different outcomes at baseline, it would seem that comparing the changes in outcome following the training would be the key outcome variable rather than comparisons of absolute values. I understand the limitations of funding, but the short duration of this study, with only 1 month observation at baseline and 3 months of observation after training, is too short to obtain enough patients for comparisons between groups leading to a significant possibility that no suggestion of differences will be observed and a subsequent, larger study will not be conducted. Since this is designed as a pilot study and the impact on mortality in such a short timeframe is unlikely, should the primary objective be a process variable rather than an outcome like mortality? Another issue with the short duration of the study is that the impact of training residents may not be immediate. If indeed their approach to patient care is different than the practice prior to training, it may take some time for the other healthcare
---

	professionals involved in trauma care to adapt to this approach. Furthermore, the impact of training the residents without providing additional training to other members of the team may limit the potential impact. Is there data on what percentage of physicians who care for trauma patients have taken either ATLS or PTC? To better understand the impact of additional training, it would help to at least know this percentage within the participating hospitals. Specific comments Abstract: At the end of the Introduction, the authors should consider stating that neither of these courses are routinely taught. It would be helpful to list the planned subgroups. Methods: Page 11, last para: it would help to state here that the residents, not faculty, are the ones who provide the initial care of trauma patients. Page 12, para 1: One or 2 units will be used for the study in each hospital. How many units are typically found in each hospital? It's not clear what refusal to participate by the residents in a specific unit would mean to the study. Page 15, para 2: Residents in the intervention arms will be asked to provide informed consent. It seems that patients in the standard care group will not be asked. Since their patients will be included in a research project, they should be asked to provide consent. Page 16, para 2: Why can't the data analysts be blinded to treatment group? Page 18, para 2: given the overall short duration of this study and the small numbers of patients expected to be enrolled at each site, the value of an interim analysis is questionable. If it is conducted, what would be the criteria for termination? The manuscript refers to the WHO trauma care checklist. What is this checklist? Is there a reference? Table 1: In the septic shock row, the use of inotropes is stated as a measurement criterion. Should this be vasopressors? Inotropes are rarely needed in sepsis. This may be a point of semantics, but it is probably more clear and more correct to use the term "non-operative" rather than "conservative" management for certain intra-abdominal injuries. Regarding the cost of treatment, isn't the issue the overall cost of care, not out-of-pocket expenses for the patient? Supplement 1: Why is male the only subgroup for outcome assessment for so many of the variables?
--	--

REVIEWER	zhang, lianyang Third Military Medical University Daping Hospital and Research Institute of Surgery Department of Cardiology
REVIEW RETURNED	26-Nov-2021

GENERAL COMMENTS	Because of the differences in the epidemiology of trauma occurrence among countries, regions and hospitals, the procedures of pre-hospital trauma treatment and in-hospital trauma treatment, the ability and level of trauma treatment vary, especially in developing countries. Trauma training is the main strategy to improve the ability of trauma treatment. Evaluating the effectiveness of training is an important basis for training and continuous improvement of training quality, but this problem has not been solved well for a long time. This article selects 6 tertiary hospitals in India that treat more than 35 cases of severe trauma per month, provide operating room, X-ray, CT and ultrasound equipment, and blood bank 24 hours service, and 2 of them are assigned to ATLS, PTC and standard treatment groups by lottery. The authors conduct prospective controlled studies to observe the impact of ATLS and PTC on patient outcomes, medical resources and investment. Although it is not truly random, the research design is relatively reasonable. It is expected that the results and conclusions of this study can further promote the development of standardized training for severe trauma treatment in developing countries. 1. As India's trauma treatment system and training situation are very characteristic, it is recommended that "India" be included in the title. 2. This research has just started and has a nearly ideal design. If there are clear results and conclusions, they should be published as soon as possible. However, there are no specific results and conclusions. It is recommended that the editorial department determine whether it is suitable for publication in a certain column of this journal.
--

VERSION 1 – AUTHOR RESPONSE

Reviewer: 1 Prof. Samuel A. Tisherman, University of Maryland School of Medicine

Comments to the Author:

General comments

- The authors have proposed a study of the effect of the Advanced Trauma Life Support (ATLS) course or the Primary Trauma Course (PTC) on patient outcomes. The impact of these courses has been difficult to test. Since these courses have not been routinely taught in India, this study is uniquely positioned to test this impact. Overall, the study is well-described and well-designed given the many difficulties and potentially confounding factors in such a study.

Thank you.

- In general, since the hospitals will likely have different outcomes at baseline, it would seem that comparing the changes in outcome following the training would be the key outcome variable rather than comparisons of absolute values.

We agree that change from baseline is an important outcome considering the heterogeneity at start. We will therefore assess and compare the primary and most secondary outcomes as both final, absolute, values and as change from baseline. This has been clarified in the 'Outcomes' section.

- I understand the limitations of funding, but the short duration of this study, with only 1 month observation at baseline and 3 months of observation after training, is too short to obtain enough patients for comparisons between groups leading to a significant possibility that no suggestion of

differences will be observed and a subsequent, larger study will not be conducted.

Thank you for this important comment. We agree that longer observation periods would be ideal and this will be an important consideration in the planning of the full-scale trial. Please note that the full-scale trial will be planned regardless of the effect-sizes identified in this pilot study, and this has now been clarified in the 'Trial Design' section.

- Since this is designed as a pilot study and the impact on mortality in such a short timeframe is unlikely, should the primary objective be a process variable rather than an outcome like mortality?

We argue that the focus on patient outcomes is what makes this research novel, as this is in our opinion what is lacking in the current evidence base for trauma life support training programs. Process measure studies are available and referenced in manuscript, see references 3,5,8,9 and 10. We will therefore keep mortality as the primary outcome, with process measures as secondary outcomes, while being cognisant of the fact that small effect sizes may be an artifact of the short observation period. However, a small effect size from this pilot study will lead to the full-scale becoming overpowered rather than underpowered.

- Another issue with the short duration of the study is that the impact of training residents may not be immediate. If indeed their approach to patient care is different than the practice prior to training, it may take some time for the other healthcare professionals involved in trauma care to adapt to this approach. Furthermore, the impact of training the residents without providing additional training to other members of the team may limit the potential impact.

We agree, and please see our previous answer in response to the short time period. We also agree with your comment on training the residents without providing additional training to other members of the team, but this is in line with how these training programs are currently being delivered in the study setting, and many other similar settings around the world. For example, the involvement of nurses during the initial management of trauma patients is very limited. We have added this to the description of the setting. The larger trial which will be of longer duration, will allow the dissemination of learnings from these courses and training of the other team members. During the pilot members of the health care staff will receive monthly briefings about the progress of the study.

- Is there data on what percentage of physicians who care for trauma patients have taken either ATLS or PTC? To better understand the impact of additional training, it would help to at least know this percentage within the participating hospitals.

We have added the approximate figures from the hospitals that have agreed to participate.

Specific comments

- Abstract: At the end of the Introduction, the authors should consider stating that neither of these courses are routinely taught. It would be helpful to list the planned subgroups.

We have added this information to the abstract.

- Methods: Page 11, last para: it would help to state here that the residents, not faculty, are the ones who provide the initial care of trauma patients.

Thank you for pointing out that this was not clear. We have made an attempt to clarify this in the 'Standard of Care' section.

- Page 12, para 1: One or 2 units will be used for the study in each hospital. How many units are

typically found in each hospital? It's not clear what refusal to participate by the residents in a specific unit would mean to the study.

There are typically six surgical units in each hospital. This information has been added to the manuscript. If residents decline to participate so that the 75% target cannot be met another unit will be selected. This has been clarified in the 'Implementation' section.

- Page 15, para 2: Residents in the intervention arms will be asked to provide informed consent. It seems that patients in the standard care group will not be asked. Since their patients will be included in a research project, they should be asked to provide consent.

The patients will provide consent. We will not collect any personal identifiable data on the residents in control group, or change their practice in any way, why we do not think that their consent is needed. This is according to ethical regulations in the study setting, and has been approved by ethical review boards at the participating hospitals. We have added this justification to the manuscript.

- Page 16, para 2: Why can't the data analysts be blinded to treatment group?

Thank you for this question. The data analysts could be blinded to the treatment group, but we did not judge the added value of this as enough considering that the purpose of this pilot study is to estimate effects for later sample size calculations and to assess the feasibility of the cluster randomized design. The data analysts will be blinded to treatment group in the full-scale trial. We have added this to the manuscript.

- Page 18, para 2: given the overall short duration of this study and the small numbers of patients expected to be enrolled at each site, the value of an interim analysis is questionable. If it is conducted, what would be the criteria for termination?

Thank you for pointing out that the criteria for termination were not included. We agree that the value of an interim analysis in the formal sense is limited, but we think that there is a value to a structured data analysis halfway through the study to assess whether it is worthwhile to continue. Reasons not to continue could include that collection of key variables, such as mortality outcomes, is unfeasible, or that patients are not consenting to data collection. We have added this to the manuscript.

- The manuscript refers to the WHO trauma care checklist. What is this checklist? Is there a reference?

Thank you for pointing out that the reference was not included. We now include the reference as a note to the table of outcomes where it is mentioned.

- Table 1: In the septic shock row, the use of inotropes is stated as a measurement criterion. Should this be vasopressors? Inotropes are rarely needed in sepsis.

Yes, it should be vasopressors. This has been revised accordingly.

- This may be a point of semantics, but it is probably more clear and more correct to use the term "non-operative" rather than "conservative" management for certain intra-abdominal injuries.

We have replaced "conservative" with "non-operative".

- Regarding the cost of treatment, isn't the issue the overall cost of care, not out-of-pocket expenses for the patient?

We have revised the this outcome from “cost of treatment” to “out-of-pocket expenditure”, as this more accurately describe what we want to assess. We will look at the cost of the treatment for the patient, and in the study setting almost all costs for trauma care are payed out-of-pocket.

- Supplement 1: Why is male the only subgroup for outcome assessment for so many of the variables?

Thank you for spotting this! There was a row missing in the table, causing what should be Yes for all patients to be Yes for men. This has been fixed.

Reviewer: 2

Prof. lianyang zhang, Third Military Medical University Daping Hospital and Research Institute of Surgery Department of Cardiology

Comments to the Author:

- Because of the differences in the epidemiology of trauma occurrence among countries, regions and hospitals, the procedures of pre-hospital trauma treatment and in-hospital trauma treatment, the ability and level of trauma treatment vary, especially in developing countries. Trauma training is the main strategy to improve the ability of trauma treatment. Evaluating the effectiveness of training is an important basis for training and continuous improvement of training quality, but this problem has not been solved well for a long time. This article selects 6 tertiary hospitals in India that treat more than 35 cases of severe trauma per month, provide operating room, X-ray, CT and ultrasound equipment, and blood bank 24 hours service, and 2 of them are assigned to ATLS, PTC and standard treatment groups by lottery. The authors conduct prospective controlled studies to observe the impact of ATLS and PTC on patient outcomes, medical resources and investment. Although it is not truly random, the research design is relatively reasonable. It is expected that the results and conclusions of this study can further promote the development of standardized training for severe trauma treatment in developing countries.

Thank you for this comment.

- As India’s trauma treatment system and training situation are very characteristic, it is recommended that “India” be included in the title.

We agree that India’s treatment system and training situation are characteristic, but not more so than the treatment system and training situation in the US, China, or many countries in Europe. Many cluster randomized controlled trials from these settings do not include the country in the title. In addition, the SPIRIT statement do not ask for the country to be included in the title. Finally, we believe that the findings of this study, and even more so the findings of the full-scale trial, will be generalisable to settings outside of India. Because of these reasons we do not include India in the title.

- This research has just started and has a nearly ideal design. If there are clear results and conclusions, they should be published as soon as possible. However, there are no specific results and conclusions. It is recommended that the editorial department determine whether it is suitable for publication in a certain column of this journal.

Thank you for this comment. Our manuscript presents a study protocol, not the results of a study.

VERSION 2 – REVIEW

REVIEWER	Tisherman, Samuel A. University of Maryland School of Medicine
REVIEW RETURNED	26-Mar-2022
GENERAL COMMENTS	The authors have responded well to the critiques.